# Validating the indicator "maternal death review coverage" to improve maternal mortality data: A retrospective analysis of district, facility, and individual medical record data

Jewel Gausman[1,2]*, Ernest Kenu[3], Richard Adanu[4], Delia A. B. Bandoh[3], Mabel Berrueta[5], Suchandrima Chakraborty[6], Nizamuddin Khan[6], Ana Langer[1], Carolina Nigri[5], Magdalene A. Odikro[3], Verónica Pingray[5], Sowmya Ramesh[6], Niranjan Saggurti[6], Paula Vázquez[5,7], Caitlin R. Williams[5,8], R. Rima Jolivet[1]

1 Women and Health Initiative, Department of Global Health and Population, Harvard University T.H. Chan School of Public Health, Boston, Massachusetts, United States of America, 2 Maternal and Child Nursing Department, School of Nursing, University of Jordan, Amman, Jordan, 3 Department of Epidemiology and Disease Control, University of Ghana School of Public Health, Accra, Ghana, 4 Department of Population, Family, and Reproductive Health, University of Ghana School of Public Health, Accra, Ghana, 5 Institute for Clinical Effectiveness and Health Policy (Instituto de Efectividad Clínica y Sanitaria (IECS)), Buenos Aires, Argentina, 6 Population Council, New Delhi, India, 7 Department of Health Science, Kinesiology, and Rehabilitation, Universidad Nacional de La Matanza, Buenos Aires, Argentina, 8 Department of Maternal & and Child Health, Gillings School of Global Public Health, University of North Carolina at Chapel Hill, Chapel Hill, North Carolina, United States of America

* jmg923@mail.harvard.edu

**Data Availability Statement:** All data have been anonymized to ensure compliance with human

## Abstract

### Background

Understanding causes and contributors to maternal mortality is critical from a quality improvement perspective to inform decision making and monitor progress toward ending preventable maternal mortality. The indicator "maternal death review coverage" is defined as the percentage of maternal deaths occurring in a facility that are audited. **Both the numerator and denominator of this indicator are subject to misclassification errors, underreporting, and bias.** This study assessed the validity of the indicator by examining both its numerator—the number and quality of death reviews—and denominator—the number of facility-based maternal deaths and comparing estimates of the indicator obtained from facility- versus district-level data.

### Methods and findings

We collected data on the number of maternal deaths and content of death reviews from all health facilities serving as birthing sites in 12 districts in three countries: Argentina, Ghana, and India. Additional data were extracted from health management information systems on the number and dates of maternal deaths and maternal death reviews reported from health facilities to the district-level. We tabulated the percentage of facility deaths with evidence of

subject protections and study protocols. The anonymized data underlying the findings are deposited here: Jolivet, R Rima; Gausman, Jewel; Adanu, Richard; Bandoh, Delia; Beruetta, Mabel; Chakraborty, Suchandrima; Kenu, Ernest; Khan, Nizamuddin; Odikro, Magdalene; Pingray, Veronica; Ramesh, Sowmya; Vázquez, Paula; Williams, Caitlin; Langer, Ana, 2022, "Validation Data for "Maternal Death Review Coverage"", https://doi.org/10.7910/DVN/3UEY4I, Harvard Dataverse, V1, UNF:6:wh1AhrYk9dDrEMsQSRgELQ== [fileUNF].

**Funding:** This work was supported by the Bill and Melinda Gates Foundation through an award to RRJ and AL (Improving Maternal Health Measurement (IMHM) Capacity and Use, grant number OPP1169546). Funders had no role in study design, data collection and analysis, decision to publish, or preparation of the manuscript.

**Competing interests:** The authors have declared that no competing interests exist.

**Abbreviations:** CRVS, civil registration and vital statistics; EPMM, Ending Preventable Maternal Mortality; HMIS, health management information system; MMR, maternal mortality ratio; MPDSR, maternal and perinatal death surveillance and response; SDG, Sustainable Development Goal; WHO, World Health Organization.

a review, the percentage of reviews that met the World Health Organization defined standard for maternal and perinatal death surveillance and response. Results were stratified by sociodemographic characteristics of women and facility location and type. We compared these estimates to that obtained using district-level data. and looked at evidence of the review at the district/provincial level. Study teams reviewed facility records at 34 facilities in Argentina, 51 facilities in Ghana, and 282 facilities in India. In total, we found 17 deaths in Argentina, 14 deaths in Ghana, and 58 deaths in India evidenced at facilities. Overall, >80% of deaths had evidence of a review at facilities. In India, a much lower percentage of deaths occurring at secondary-level facilities (61.1%) had evidence of a review compared to deaths in tertiary-level facilities (92.1%). In all three countries, only about half of deaths in each country had complete reviews: 58.8% (n = 10) in Argentina, 57.2% (n = 8) in Ghana, and 41.1% (n = 24) in India. Dramatic reductions in indicator value were seen in several subnational geographic areas, including Gonda and Meerut in India and Sunyani in Ghana. For example, in Gonda only three of the 18 reviews conducted at facilities met the definitional standard (16.7%), which caused the value of the indicator to decrease from 81.8% to 13.6%. Stratification by women's sociodemographic factors suggested systematic differences in completeness of reviews by women's age, place of residence, and timing of death.

## Conclusions

Our study assessed the validity of an important indicator for ending preventable deaths: the coverage of reviews of maternal deaths occurring in facilities in three study settings. We found discrepancies in deaths recorded at facilities and those reported to districts from facilities. Further, few maternal death reviews met global quality standards for completeness. The value of the calculated indicator masked inaccuracies in counts of both deaths and reviews and gave no indication of completeness, thus undermining the ultimate utility of the measure in achieving an accurate measure of coverage.

## Introduction

Sustainable Development Goal (SDG) 3.1 calls for reducing the global average maternal mortality ratio (MMR) to <70 maternal deaths per 100,000 live births by 2030. To achieve this goal, most countries must reduce their national MMR by at least two-thirds; those with very high maternal mortality will need to reduce their MMR even further [1]. In 2017, the global average MMR was 211/100,000 [2], requiring an average 7.6% annual reduction in MMR to reach the SDG global target [3].

Accurate measurement of maternal mortality is critical to inform high-level decisions about programming, practice, and resource allocation and to monitor progress in order to prevent maternal deaths [4, 5]. However, half of World Health Organization (WHO) global member states do not have a functioning civil registration and vital statistics (CRVS) system capable of capturing data on maternal deaths and causes; thus, it is estimated that half to two-thirds of all deaths are not registered [6, 7]. Incomplete reporting of maternal deaths to formal registration systems hampers efforts to understand and address the underlying causes [8].

Maternal and perinatal death surveillance and response (MPDSR) is another approach to track maternal deaths and learn from them to prevent future mortality [1, 4]. A 2016 WHO

survey of MPDSR implementation indicated that a majority of low- and middle-income countries have only partially implemented MPDSR programs at national scale [8, 9]. Ideally, national CRVS systems would be complemented by universal maternal death review programs, whose data would be linked to ensure that every death is counted, its root causes understood, and lessons translated into effective programming and resource allocation to end preventable maternal mortality in each country [10, 11].

The WHO MPSDR Technical Working Group released guidance describing the purpose and essential components of a complete maternal death review [9, 12]. This guidance stipulates that each maternal death occurring in a facility should undergo an organized committee review that meets minimum standards (**Box 1**).

## Box 1. MPSDR defined standards for complete maternal death review

- Establishes medical cause of death

- Determines if confirmed maternal death

- Determines nonmedical factors related to the death

- Assesses quality of medical care

- Determines if death was avoidable

- Provides recommendations for immediate action

- Compiles investigations and sends them to district level with recommendations for action

Complete review of deaths is important to provide evidence of problems in service delivery and quality of care, and to highlight areas for improvement [13]. Health care decision makers are limited in their efforts to end maternal preventable deaths if they do not have comprehensive information on the number and causes of such deaths [14]. Thus, maternal death review coverage is an important indicator to track. At the launch of the SDGs, WHO published "Strategies toward Ending Preventable Maternal Mortality" [1], which is the global strategic framework for maternal health and survival during the SDG period. "Maternal death review coverage" is a core indicator in this monitoring framework, prioritized by stakeholders for its potential to advance efforts toward EPMM Key Theme 5: "Improve metrics, measurement systems, and data quality to ensure that every maternal and newborn death is counted" [15]. The indicator is defined as the percentage of maternal deaths occurring in a facility that were audited [16].

**While critical for ending preventable maternal deaths, the indicator measuring "**maternal death review coverage" **is subject to significant measurement challenges. Both its numerator and denominator are subject to misclassification errors, underreporting, or bias [17]. Some challenges are specific to the numerator. For example, death reviews may be of variable quality, introducing the danger that some reviews included in the numerator reflect only a ticked box and do not serve their intended purpose [18].** In recent years, the concept of "effective coverage" has gained traction in the field of maternal newborn health

with the understanding that to be meaningful, measures of coverage should go beyond a simple frequency or tally to also account for quality or comprehensiveness [19–20]. Thus, a simple count of reported maternal death reviews, if they do not meet standards for quality and completeness, may not yield a valid measure of the phenomenon that decisionmakers seek to track. **Further, there may be systematic differences in the "missingness" of maternal deaths that undergo review, if certain types of maternal deaths are not reviewed based on factors that form the basis for inequities, such as women's sociodemographic characteristics [22–24].**

## There are also specific challenges related to the denominator of the indicator

Even with well-functioning systems for capturing maternal deaths, some deaths may be missed or misclassified. Certain types of maternal deaths may be more likely to fall into this category, for example deaths that occur outside of health facilities [25, 26]. **Lack of continuity of care across settings and lack of data sharing mean that deaths that occur outside of facilities (in communities or in transit between facilities) may not be captured in death registration systems at facility or subnational administrative levels [27].** Deaths attributable to certain causes may be subject to differential incomplete reporting **due to factors such as stigma related to the cause of death itself, such as abortion-related deaths, or because they are mistakenly classified as non-maternal deaths, such as later-occurring maternal deaths between >42 days and <1 years after pregnancy termination [22, 27]. Some facility deaths may not be accurately captured because of poor quality reporting and a lack of certification of the causes of death [28, 29] or due to fear over reprisal, shame, and blame [30].** Likewise, it is reasonable to assume that deaths of women from certain vulnerable populations may be less likely to be registered and thus monitored effectively, suggesting that comprehensive maternal death registration is also important for achieving equity.

This study sought to validate the indicator "maternal death review coverage" by assessing both its numerator and denominator and comparing different sources of data. The first aim was to assess the numerator, i.e., the magnitude of maternal death review coverage, considering the quality and completeness of each recorded review to assess "effective coverage" of maternal death reviews in the study districts. This was based on comparison of reported review coverage to objective evidence of the number and completeness of maternal death reviews that took place as a proportion of all reported facility-based maternal deaths. The second aim was to assess the denominator, i.e., to validate the number of facility-based maternal deaths reported in study health facilities within each district, via comparison with facility health management information systems (HMIS) and chart data. Finally, we explored how the value of the reported indicator changed based on differing numerators and denominators obtained from district- versus facility-level data.

## Methods

This study retrospectively reviewed primary data collected from charts and records at health facilities combined with data extracted from HMIS at the district/provincial level.

### Study setting

This study took place in four subnational geographic areas in Argentina (provinces of Buenos Aires Region V, Jujuy, La Pampa, and Salta), Ghana (Bunkpurugu Yunyoo and Tolon in the Northern Regions; Techiman and Sunyani Municipal in the Brong-Ahafo Region), and India (Gonda and Meerut in the state of Uttar Pradesh; Krishnagiri and Thirvallu in the state of Tamil Nadu). The three countries included in this study were selected through a competitive

process with the intention of representing one country from Africa, Asia and Latin America/ Caribbean based on World Bank classification. Subnational districts (India and Ghana) and provinces (Argentina) were selected purposefully by way of a two-staged sampling strategy based on a composite index of key maternal health indicators used as a proxy to measure health system performance. One high performing and one low performing state/region in each country were selected for inclusion, as well as one high performing and one low performing district/province within each state were selected. More details on the selection for each study setting are available in the study protocol [31].

## Data collection and sources

Data collection took place from March 2020 to March 2021 and was led by collaborating institutions based in each study country, which were familiar with the local research context. Further, each institution and research team member had extensive experience conducting human subjects research in their respective countries. Data collectors in each country were comprised of clinicians and individuals with advanced training in public health. Data collectors underwent extensive training prior to data collection. Training of data collectors in each country included an overview to the study, research regulations and protocol, how to conduct an interview, field work procedures, data quality, and a detailed review of all data collection tools and procedures.

## Facility data

In Argentina and India, all public facilities where births took place were included in primary data collection. Private facilities were not included in Argentina and India as there were no registered private facilities that reported their data on maternal deaths to the government. In Ghana, all private-registered and public facilities where births took place were included, as private-registered facilities reported data on maternal deaths to the government. Eligible facilities were categorized in each country according to primary, secondary, and tertiary, based on a scoping review and synthesis of available facility acuity frameworks, to standardize them across countries. Our facility acuity definitions, with reference citations, are available in **S2 File**.

All eligible facilities were visited by a trained team of researchers who reviewed medical charts and patient records, hospital registers, and maternal morbidity and mortality review committee reports and records to determine the number of deaths evidenced at the facility during the previous 12 months and to document the completeness of death reviews. If a maternal death was documented in any of these sources, it was recorded as providing evidence of a maternal death at that facility.

Data on all maternal deaths that occurred in any eligible facility during the study period were recorded. Data collected on each maternal death included the date of death, woman's age at death, woman's place of residence (rural or urban), and timing of death (during pregnancy, during delivery, during or within 42 days of abortion, or within 42 days of delivery). In Argentina, for feasibility reasons, the count of maternal deaths within each study province was derived from Argentina's Civil Registration and Vital Statistics (CRVS) system, which has been classified as type A/type I based on high quality of the data [32, 33].

In addition to collecting data on each maternal death, data on all death reviews that were conducted at eligible facilities were also collected. Data collected on death reviews included death date, review date, and type of documentation that supported the review (meeting minutes, meeting report, or other). The study team further reviewed the content and recorded whether the documentation available substantiated that the review satisfied the seven criteria outlined in the definitional standard (see Box 1). Content was reviewed for compliance with

the WHO technical guideline, this study's reference standard, as it is the most universally applicable and universally recognized reference standard available [9, 12, 34].

Data collected from facilities at all sites used a standard paper-based data collection form with a separate question for each of the variables identified above. All maternal deaths and death reviews were documented individually on the form. One form was used for each facility.

### District/provincial data

To obtain data on maternal deaths reported to the government from eligible facilities, we extracted from HMIS on all maternal deaths that were reported during the study period. We also extracted the facility identifiers where each death was recorded as occurring, the date of each death, and whether a death had a corresponding death review reported as having been conducted, and the type of documentation (meeting minutes, meeting report, or other) that was provided in support of review. Data extracted from the HMIS system were recorded on a standard paper-based form. One form was used for each facility so that it could be easily matched with the data extracted from facility-records.

### Data management and quality

All study data were entered into RedCap according to a standardized format. All data that were entered into RedCap underwent extensive review by supervisory members of each the study team to ensure accuracy. Study data were managed using REDCap electronic data capture tools hosted at the Harvard Medical School [35, 36]. REDCap (Research Electronic Data Capture) is a secure, web-based software platform designed to support data capture for research studies, providing 1) an intuitive interface for validated data capture; 2) audit trails for tracking data manipulation and export procedures; 3) automated export procedures for seamless data downloads to common statistical packages; and 4) procedures for data integration and interoperability with external sources.

### Data analysis

**Outcome definitions.** All maternal deaths that were evidenced at an eligible facility were tabulated, as well as all maternal deaths reported to the district/provincial level through the HMIS system. We coded a death review as being complete if it satisfied each of the seven criteria defined in the MPSDR standards for complete maternal death review outlined in Box 1. If it did not satisfy all seven criteria, then it was considered an incomplete review.

**Analytic approach.** To validate the indicator, we varied the indicator's numerator and denominator and explored the differences in the indicator's value.

We began by validating the numerator's indicator, which was defined as the number of maternal deaths that occurred at a facility that were reviewed. To validate the numerator, we operationalized it in three different ways using different sources of data:1) the number of maternal deaths that occurred in a facility that had evidence of a review, 2) the number of maternal deaths that occurred in a facility that had a review that met the WHO MPDSR standard, and 3) the number of maternal deaths with reviews that were reported to the district/provincial level.

We also explored implications related to equity in relation to the numerator. To do so, we descriptively compared sociodemographic characteristics of women (age, place of residence, and timing of death) for which reviews were and were not conducted, as well as the percentage of complete and incomplete reviews according to facility-level characteristics (facility location and type of facility). The socio-demographic variables included in the study were based on the EPMM Equity Stratifiers and described in the study's published protocol (16, 32). These

variables were selected because they are routinely available in clinic records in cases of maternal death, and allowed us to disaggregate our data by specific subpopulations facing social disadvantage. Other variables that would allow us to assess equity and discrimination, such as the woman's educational attainment, wealth, and profession, were not available in retrospective clinical data.

The indicator's denominator is defined as all maternal deaths that occurred in a facility within a given geographical unit. To verify the denominator, we varied the data source from which the counts of facility-based maternal deaths were obtained. We derived the count of maternal deaths from two different sources 1) the total number of maternal deaths evidenced at health facilities, and 2) the total number maternal deaths reported to the district/provincial level.

We compiled the total number of deaths that occurred at each facility found through record review and compared it to the total number of deaths reported to the district or provincial level in the HMIS. To do this, we traced individual deaths from the facility by date of death and facility identifiers, including facility name and location, to match the facility record with data on individual deaths extracted from the HMIS. As maternal death is a relatively rare outcome, there were no facilities in any of the three countries that had more than one maternal death occur on a given date. The matching procedure enabled us to assess accuracy of the reported data in the event of a mismatch between the date recorded at a facility and the date recorded in the HMIS. Any mismatches found in the date of death between individual facility records and what was recorded in the HMIS were considered an important finding that enabled us to assess reporting accuracy. Any such cases were further verified and investigated by the research team at the facility and district level.

To examine equity in relation to the denominator, we compared the maternal deaths that were reported to the district/provincial level versus those not reported by facility location (urban or rural) and type of facility (primary, secondary, or tertiary), as women's individual sociodemographic information was not available in the HMIS system.

In the final step, we varied construction of the indicator to explore differences in its value. We calculated the indicator three different ways reflecting the different data sources available for both the numerator and denominator as detailed in Table 1. Note that due to the rarity of maternal death, we only present raw number of deaths in each subnational study area if there were >15 maternal deaths over the study period to reduce risk of indirect identification of women whose maternal death was being analyzed.

## Ethics approval

**The Institutional Review Board (IRB) of the Harvard T.H. Chan School of Public Health approved this study on 4 September 2019 (**approval ID: IRB19-1086**). The research is classified as Level 4 Data using Harvard's Data Security Policy.** The study also was approved in Argentina by the Comité de Ética de la Investigación de la Provincia de Jujuy (approval ID not applicable), Comisión Provincial de Investigaciones Biomédicas de la Provincia de Salta

**Table 1. Indicator definitions for "maternal death review coverage".**

|  | Indicator Definition 1 | Indicator Definition 2 | Indicator Definition 3 |
|---|---|---|---|
| **Numerator** | Number of maternal deaths documented at a facility that have evidence of a review | Number of maternal deaths documented at a facility that have evidence of a complete review | Number of maternal death reviews reported to district/province |
| **Denominator** | Number of maternal deaths evidenced at facilities | Number of maternal deaths evidenced at facilities | Number of maternal deaths reported to district/province |
| **Data Source** | Facility | Facility | HMIS |

(approval ID: 321-284616/2019), Consejo Provincial de Bioética de la Provincia de La Pampa (approval ID not applicable), and Comité de Ética Central de la Provincia de Buenos Aires (approval ID: 2919-2056-2019); in Ghana by the Ghana Health Service Ethical Review Board (approval ID: GHS-ERC022/08/19); and in India by the national population council IRB (approval ID: 889) and local Sigma-IRB (approval ID: 10052/IRB/19-20).

Patient consent was not required by any of the seven ethical review boards that reviewed the full study methods and instruments because this was retrospective review of health data.

## Results

### Distribution of maternal deaths evidenced in facilities

Facilities included in the study are detailed in **S2 File** **and include** 34 facilities in Argentina, 51 facilities in Ghana, and 282 facilities in India. **Table 2** describes the maternal deaths that were evidenced from facility records in each country: 17 deaths in Argentina, 14 deaths in Ghana, and 58 deaths in India. In all countries, most maternal deaths occurred in tertiary facilities: 88.2% (n = 15) in Argentina, 64.3% (n = 9) in Ghana, and 65.5% (n = 38) in India.

### Completeness of maternal death reviews

Audit findings detailing the content of each maternal death review conducted by definitional component across study countries are displayed in **Fig 1**. In all three countries, 100% of death reviews conducted established the medical cause of death. The majority of reviews conducted in all countries also confirmed that the death was a maternal death (76.5% in Argentina (n = 13), 100% in Ghana (n = 14), and 82.8% in India (n = 23). Determining if the death was avoidable was the least common component in death reviews, with only 42.8% of death reviews in India (n = 12) and 64.7% in Argentina including this component (n = 11). In Ghana, the least commonly observed components were determining any non-contributing medical factors (57%, n = 8) and assessing the quality of care received (57.0%, n = 8).

**Table 3** shows that in both Argentina and India, approximately 17% of maternal deaths had no evidence that a death review had occurred, while in Ghana, all deaths had at least some form of death review conducted. Overall, >80% of deaths had evidence that a death review had been conducted, including both complete and incomplete reviews. In India, a much lower percentage of deaths occurring at secondary-level facilities (61.1%) had evidence that any death review had been conducted compared to deaths in tertiary-level facilities (92.1%). In all three countries, only about half of deaths in each country had complete reviews: 58.8% (n = 10) in Argentina, 57.2% (n = 8) in Ghana, and 41.1% (n = 24) in India.

**Table 2. Number of maternal deaths that occurred in facilities and distribution by location and facility type in Argentina, Ghana and India.**

| | Deaths Evidenced at Facilities | | |
| --- | --- | --- | --- |
| | **Argentina** | **Ghana** | **India** |
| *Facility Location* | | | |
| Urban | 17 (100.0) | 14 (100.0) | 38 (65.2) |
| Rural | 0 (0.0) | 0 (0.0) | 20 (34.5) |
| *Facility Type* | | | |
| Primary | 0 (0.0) | 0 (0.0) | 2 (3.5) |
| Secondary | 2 (11.8) | 5 (35.7) | 18 (31.0) |
| Tertiary | 15 (88.2) | 9 (64.3) | 38 (65.5) |
| Total | 17 (100.0) | 14 (100.0) | 58 (100.0) |

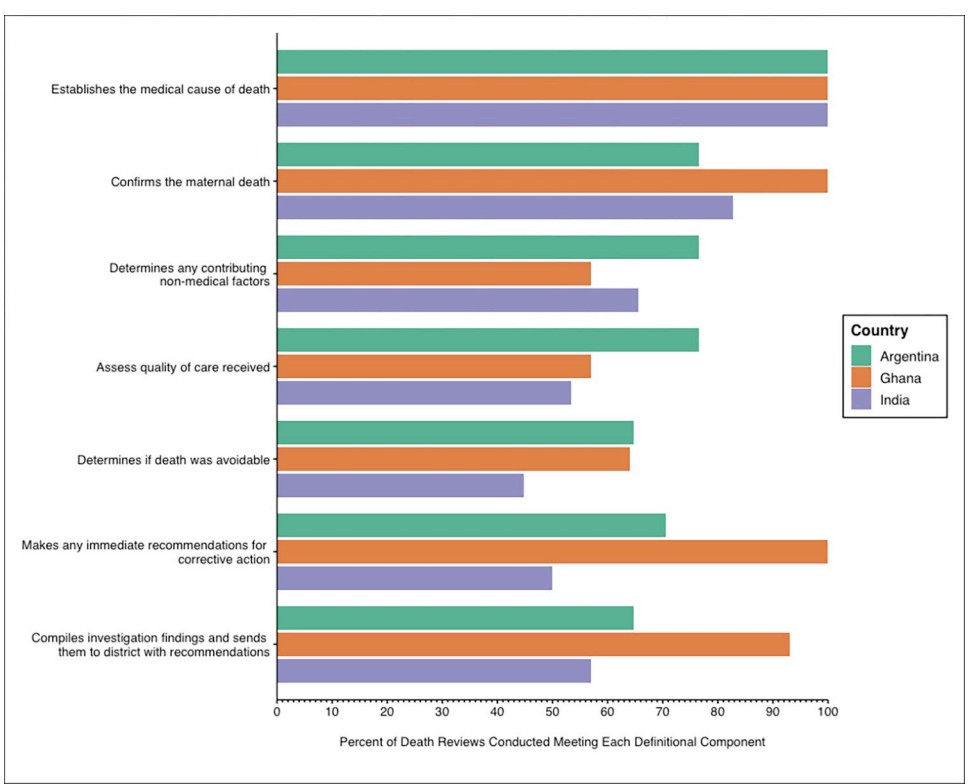

**Fig 1. Percent of death reviews in Argentina, Ghana, and India containing each defined component of a complete review as outlined by the World Health Organization maternal and perinatal death surveillance and response technical working group.**

Characteristics of the women whose deaths had complete versus incomplete reviews are in **Table 4**. Incomplete death reviews were more frequent among women from rural areas. In India, 70.3% (n = 29) of deaths among women from rural areas were incomplete compared to only 29.4% (n = 5) in urban areas. In Ghana, all (n = 3) of death reviews conducted on women who lived in rural areas were incomplete compared to 27.3% (n = 3) in urban areas.

**Table 3. The distribution of maternal deaths with no review, some evidence of a review, and complete reviews by facility location and type in Argentina, Ghana, and India.**

|  | Argentina n (%) | | | Ghana n (%) | | | India n (%) | | |
|---|---|---|---|---|---|---|---|---|---|
|  | Percent of Deaths with no Evidence of a Review | Percent of Deaths with Any Review | Percent of Deaths with Complete Review | Percent of Deaths with no Evidence of a Review | Percent of Deaths with Any Review | Percent of Deaths with Complete Review | Percent of Deaths with no Evidence of a Review | Percent of Deaths with Any Review | Percent of Deaths with Complete Review |
| Facility Location |  |  |  |  |  |  |  |  |  |
| Urban | 3 (17.7) | 14 (82.3) | 10 (58.8) | 0 (0.0) | 14 (100.0) | 8 (57.2) | 3 (7.9) | 35 (92.1) | 21 (55.3) |
| Rural | 0 (0.0) | 0 (0.0) | 0 (0.0) | 0 (0.0) | 0 (0.0) | 0 (0.0) | 7 (35.0) | 13 (65.0) | 3 (15.0) |
| Facility Type |  |  |  |  |  |  |  |  |  |
| Primary | 0 (0.0) | 0 (0.0) | 0 (0.0) | 0 (0.0) | 0 (0.0) | 0 (0.0) | 0 (0.0) | 2 (100.0) | 1 (50.0) |
| Secondary | 1 (50.0) | 1 (50.0) | 0 (0.0) | 0 (0.00 | 5 (100.0) | 0 (0.0) | 7 (38.9) | 11 (61.1) | 2 (11.2) |
| Tertiary | 2 (13.3) | 13 (86.7) | 10 (66.7) | 0 (0.0) | 9 (100.0) | 8 (88.9) | 3 (7.9) | 35 (92.1) | 21 (55.26) |
| Total | 3 (17.7) | 14 (82.3) | 10 (58.8) | 0 (0.0) | 14 (100.0) | 8 (57.2) | 10 (17.2) | 48 (82.8) | 24 (41.4) |

**Table 4. Sociodemographic characteristics of women with complete vs. incomplete death reviews in Argentina, Ghana, and India.**

| | Argentina n (%) | | Ghana n (%) | | India n (%) | |
|---|---|---|---|---|---|---|
| | Complete | Incomplete | Complete | Incomplete | Complete | Incomplete |
| Reviews (n) | 10 | 7 | 8 | 6 | 24 | 34 |
| *Age at Death* | | | | | | |
| 15–24 years | 1 (100.0) | 0 (0.0) | 2 (100.0) | 0 (0.0) | 13 (31.8) | 13 (61.9) |
| 25–34 years | 2 (50.0) | 2 (50.0) | 3 (42.9) | 4 (57.9) | 14 (43.8) | 18 (56.3) |
| 35+ years | 7 (58.33) | 5 (41.7) | 3 (60.0) | 2 (40.0) | 2 (40.0) | 3 (60.0) |
| *Woman's Place of Residence* | | | | | | |
| Urban | 1 (100.0) | 0 (0.0) | 8 (72.7) | 3 (27.3) | 12 (70.6) | 5 (29.4) |
| Rural | 6 (37.5) | 10 (62.5) | 0 (0.0) | 3 (100.0) | 12 (29.3) | 29 (70.3) |
| *Timing of Death* | | | | | | |
| During Pregnancy | 0 (0.0) | 1 (100.0) | 4 (80.0) | 1 (20.0) | 7 (35.0) | 13 (65.0) |
| During Delivery | -- | -- | 2 (66.7) | 1 (33.3) | 2 (33.3) | 4 (66.7) |
| During or Within 42 Days After Abortion | 1 (100.0) | 0 (0.0) | -- | -- | 1 (100.0) | 0 (0.0) |
| Within 42 Days After Delivery | 9 (60.0) | 6 (40.0) | 2 (33.3) | 4 (66.7) | 14 (45.2) | 17 (54.8) |

In terms of the timing of maternal death, among women in Ghana who died within 42 days after delivery, only 33.3% (n = 2) had a complete review. Conversely, in both Ghana and India, the majority of women who died during pregnancy had complete death reviews (80.0% (n = 4) in Ghana; 65.0% (n = 13) in India).

In Argentina, as a large majority of maternal deaths occurred among older women, women from rural areas, and within 42 days after delivery, there was little opportunity to assess completeness of reviews according to sociodemographic characteristics.

## Reporting of maternal deaths and death reviews to district

Data on the deaths and reviews reported to the district level from facilities are in **Table 5**. The percentage of deaths evidenced at facilities that were reported to the district-level was 100.0% in Argentina (n = 17), 64.3% in Ghana (n = 9), and 87.9% in India (n = 51). In Argentina and India, only 82.3% (n = 14) and 79.3% (n = 46) death reviews that were conducted at facilities were ultimately reported to the district-level. Only in Ghana did 100% of maternal deaths have evidence of a review, yet only 57% were complete. At the district level, 64% of maternal deaths evidenced at facilities were reported and 64% of reviews were reported in Ghana. Facility data

**Table 5. Maternal deaths and maternal death reviews reported to the district-level from facilities in Argentina, Ghana and India.**

| | Argentina | | Ghana | | India | |
|---|---|---|---|---|---|---|
| | Deaths Evidenced at Facilities and Reported to District | Deaths Evidenced at Facilities with Reviews Reported to District (n) | Deaths Evidenced at Facilities and Reported to District | Deaths Evidenced at Facilities with Reviews Reported to District (n) | Deaths Evidenced at Facilities and Reported to District | Deaths Evidenced at Facilities with Reviews Reported to District (n) |
| *Facility Location* | | | | | | |
| Urban | 17 (100.0) | 14 (82.3) | 9 (64.3) | 9 (64.3) | 35 (92.1) | 35 (94.6) |
| Rural | -- | -- | -- | -- | 16 (80.0) | 11 (55.0) |
| *Facility Type* | | | | | | |
| Primary | -- | -- | -- | -- | 2 (100.0) | 2 (100.0) |
| Secondary | 2 (100.0) | 1 (50.0) | 3 (60.0) | 4 (80.0) | 14 (77.8) | 11 (61.1) |
| Tertiary | 15 (100.0) | 13 (86.7) | 6 (66.7) | 5 (55.6) | 35 (92.1) | 33 (86.8) |
| Total | 17 (100.0) | 14 (82.3) | 9 (64.3) | 9 (64.3) | 51 (87.9) | 46 (79.3) |

from Argentina and India indicate that not all maternal deaths were reviewed at the facility level, and an even smaller proportion had complete reviews. However, in both Argentina and India, most maternal deaths and reviews that were conducted (regardless of completeness) were reported to the district.

## Comparing the value of the indicator obtained from different data sources and measurement approaches

Table 6 presents components of the indicator calculations for different definitions of "maternal death review coverage" (numerators and denominators) as well as the calculations aggregated across the four subnational study areas in each country. In all countries, the indicators calculated as the percentage of maternal deaths documented at facilities that had evidence of a review (Indicator Definition 1) and the percentage of maternal deaths reported to the district with a review reported to the district (Indicator Definition 3) yielded similar results. However, when comparing components of the indicator, there were some important differences across countries. In Ghana, the value of the indicator was the same but was based on incomplete reporting of maternal deaths and reviews to the district level. In all countries, the value of the indicator dropped substantially when factoring in the quality of death reviews—hovering around 50% effective coverage of maternal death reviews in all countries.

Fig 2 highlights variation in the value of the "maternal death review coverage" indicator based on its formulation across subnational geographic study areas with evidence of a maternal death. Of the subnational geographic study areas, there was evidence of >15 maternal deaths occurring during the study period in Gonda and Meerut in India. In each district, the value of the indicator as calculated based on facility data (Indicator Definition 1) and district data (Indicator Definition 3) were generally very similar, although the value based on district rather than facility data was slightly higher in Meerut (80.8% to 78.6%) and Gonda (88.2% to 81.8%). The reason for this difference is disproportionate underreporting in the numerator (number of reviews reported to districts) and denominator (number of maternal deaths reported to districts) when comparing facility and district data. In Gonda, there were 22 maternal deaths verified at facilities, and 18 of those deaths had evidence of a review, thus resulting in a facility-level coverage estimate of 81.8%. However, 77.7% of maternal deaths verified at facilities were reported to the district (17 of 22 maternal deaths), while 83.3% of reviews evidenced at facilities were reported to the district (15 of 18 reviews), resulting in a coverage estimate of 88.2%. Similarly, in Meerut, there were 28 deaths verified at facilities, and 22 of those deaths had evidence of a review, resulting in a coverage estimate of 78.6% (22 reviews among 28 maternal deaths). At the district level, 92.9% of maternal deaths were reported (26 of 28 deaths), but 95.5% of reviews were reported (21 of 22 reviews), resulting in a coverage estimate of 80.8%.

The largest reduction in value of the indicator occurred when incorporating the quality of death reviews based on whether their content met the defined standard. Dramatic reductions in indicator value were seen in several subnational geographic areas, including Gonda and

**Table 6. "Maternal death review coverage" indicator calculations by country.**

| Country | Indicator Definition 1 | Indicator Definition 2 | Indicator Definition 3 |
|---|---|---|---|
| | # of Maternal Deaths with Evidence of Review/# of Maternal Deaths Evidenced at Facilities | # of Maternal Deaths with Complete Review/# of Maternal Deaths Evidenced at Facilities | # of Maternal Death Reviews Reported to District/# of Maternal Deaths Reported to District |
| Argentina | 14/17 = 82.4% | 10/17 = 58.8% | 14/17 = 82.4% |
| Ghana | 14/14 = 100% | 8/14 = 57% | 9/9 = 100% |
| India | 48/58 = 82.8% | 24/58 = 41.4% | 46/58 = 79.3% |

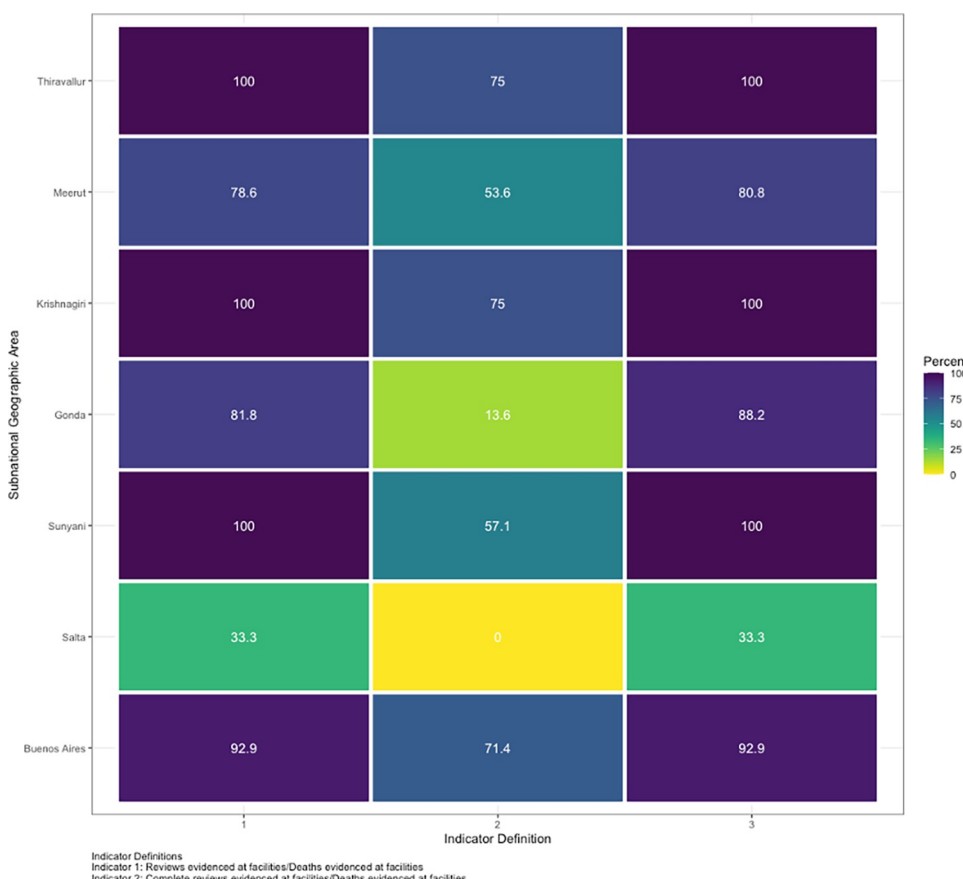

**Fig 2. Variation in value of maternal death review coverage indicator calculated using facility vs. district data.**

Meerut in India and Sunyani in Ghana. For example, in Gonda only three of the 18 reviews conducted at facilities met the standard (16.7%), causing the value of the indicator as calculated using facility-level data to decrease from 81.8% to 13.6%. In Salta, the value of the indicator decreased from 33.3% to 0% when considering only reviews that meet the standard.

## Discussion

Meaningful measurement of maternal death review coverage depends on valid estimates of both maternal deaths and death reviews. Our results raise important concerns about whether simply counting the number of evidenced reviews in a facility fully reflects coverage in the most meaningful way. Further, our results show important problems related to incomplete reporting of both deaths and reviews from the facility to the district that may undermine the indicator's estimates. The idea of effective coverage is usually applied to coverage of care interventions [19, 20], but our results support the contention of Jannati et al. [21] that this conceptual framework is relevant for other measures of coverage addressing the health system and policy arena where a simple count does not capture important information on content and quality.

The relatively small changes we observe in coverage estimates when we compare those obtained from using facility- or district-level data, before accounting for review quality, could be interpreted as showing that maternal death review coverage can be reliably calculating

using only district-level data, which would allow MDPSR programs to calculate the indicator more efficiently. However, our results show that the similarity of estimates obtained from the facility versus district-level is artefactual. Validation against primary facility-level data revealed substantial underreporting of the number of maternal deaths that occurred in facilities to the district. Taking the results from Ghana as an example, coverage estimates calculated using both facility- and district-level data suggest that 100% of deaths were reviewed. However, more than half the reviews conducted at facilities and half of the maternal deaths that occurred at facilities went unreported to the district. Thus, while these estimates produce the same value due to congruence in underreporting both the numerator and denominator, the indicator calculated using district-level data represents inaccurate and incomplete reporting. In Ghana, records of death reviews are often reported directly to the regional level, which is one level higher than the district-level; therefore, some facilities may bypass the district-level when they report maternal deaths, perhaps explaining some of the incomplete reporting observed. This highlights that any global indicator seeking to measure maternal death review coverage using data within the health system, rather than that obtained from direct facility audits, would need to clearly specify the most appropriate level of the health system from which to abstract data, which may vary across countries. In other study areas, data reflect a different phenomenon at the subnational level, whereby maternal deaths that are reviewed appear more likely to be reported to the district. Our results from Meerut and Gonda in India illustrate this concern—disproportionate reporting of the numerator and denominator from the facility to the district level in these areas resulted in an observed increase in coverage calculated for the indicator using district-level data.

Given these results, our study raises concerns over accuracy of component values of the measure, despite relative consistency in the summary statistic for the indicator value calculated using district or facility data. In another setting, it would not be difficult to see how district-versus facility-level estimates could vary from each other on much greater scale than observed in our study due to similar inconsistencies in reporting. Given that maternal death is both a rare and severe event, accuracy of each count in both the numerator and denominator of this indicator is important. Each maternal death should be accounted for and learned from. Future research is warranted to explore whether reporting inconsistencies are widespread in other settings.

In terms of health equity, stratification by women's sociodemographic factors points to the possibility of systematic differences in the completeness of reviews according to a woman's age, place of residence, and timing of her death; however, our ability to assess these differences is limited given the small number of maternal deaths in our study. Further research is warranted to explore whether these observations are present on a larger scale and across other countries. Nonetheless, such differences would be masked by only calculating the indicator in aggregate, without stratifying to explicitly examine issues of equity, as recommended in the indicator framework for EPMM [15].

We believe that effective coverage, operationalized as coverage of death reviews that meet a defined reference standard, is central to the indicator's construct validity. For example, reviews that are incomplete or of poor quality do little to provide the system with the necessary information to effectively learn from maternal deaths. Our results show that the use of only facility or district records to calculate the indicator, without auditing death review content, has limited validity in relation to the underlying construct given dramatic reductions in the indicator value when factoring in the quality of death reviews. Previous research conducted in 46 African countries found that only half had guidelines for maternal death review content, and only 40% of those countries had implemented those guidelines [37]. The maternal death review guidelines we identified in each of our study settings include some but not all elements of the

current global standards for essential content [38–40]; ensuring the existence of national guidelines for maternal death reviews with content that matches global standards for quality, as well as ensuring their reliable implementation, is needed to support improved validity of the indicator. In Argentina, systematic efforts were made during the last decades to improve active reporting, reduce underreporting, and conduct process of care analyses of maternal deaths, including training on how to conduct reviews based on root cause analysis, the three delays approach and other analysis methodologies proposed by the Pan-American Health Organization [38]. However, Argentina has not adopted the WHO criteria for classifying each review as complete, and there is no history of surveillance of the quality of the reviews. The Government of India has launched guidelines to carry out maternal death review. Reporting of maternal deaths has improved over the years, but there is great interstate variability in reporting [41]. According to government guidelines all maternal deaths should be reviewed. For deaths occurring at home, in communities and in facilities, information should be gathered and supplied to an oversight committee. The guideline clearly describes the stakeholders responsible; however, implementation of those guidelines on the ground still needs improvement.

This study is subject to several strengths and limitations. A strength of this study is that we verified all maternal deaths, maternal death reviews, and content of reviews by collecting primary data from birth facilities, and we cross-validated facility-level data with data reported to districts. In terms of limitations, the number of deaths included in our studies was quite small as maternal death is a rare event. Therefore, our ability to draw statistical inferences from our data is limited and we report numerical, qualitative differences only. We can also only generalize to the geographic units in which our study took place, as national-level generalizability is not the purpose of this study. Further, we traced deaths by recorded date of death from facilities to districts. While our study team examined individual medical records to determine the number of deaths evidenced in facilities, it is possible that some maternal deaths were overlooked in facility records. Additionally, there may be some inaccuracy in the dates of death recorded in facility or district records. While we attempted to triangulate across all available records to match deaths in facilities that occurred on similar dates to those reported to the district, if records were missing either at the facility or the district, it is possible that errors in record keeping meant that not all deaths could be traced. This is both a limitation to the study and an important finding. Finally, the indicator validated through our research was limited to death review coverage among facility-based maternal deaths at facilities designated as birthing facilities. As such, our study results provide no information on the coverage of reviews for maternal deaths that did not occur in such a facility. However, we found no evidence of maternal deaths reported at the district level but not linked to a facility in any of our research settings. In Argentina and India, this is important given that private facilities were not included in the study, which could have an impact on both the numerator and the denominator as death reviews may be less frequent in facilities that are not part of the public hospital network; however, as private facilities in these countries do not report maternal deaths to the government, HMIS data were unavailable from such facilities, which would have limited our ability to compare the value of the indicator using different data sources. In Ghana, there were no maternal deaths found at any facilities in three of the four subnational study areas. This may be partially explained by Ghana's policy of referring high-risk women to higher-level facilities, of which there were none in the study areas where no deaths were evidenced. While experience demonstrates that community-based maternal deaths occur, including deaths in transport, as well as deaths at lower level facilities that do not typically handle deliveries, our results suggest that such deaths may not be reliably reported to the district level in all three countries [42]. This finding has implications for both future research and death registration policy and practice, especially in terms of ensuring that all maternal deaths are both counted and reviewed.

## Conclusion

Our study assessed the validity of an important indicator for ending preventable deaths—coverage of reviews of maternal deaths occurring in facilities—in three study settings. We found incompleteness in maternal deaths recorded at birth facilities and reported to districts from facilities; notably, we found that very few maternal death reviews met global quality standards for completeness of review information. Further, the value of the calculated indicator masked inaccuracies in counts of both deaths and reviews that were validated through primary data review and gave no indication of the completeness, and thus ultimate utility, of reported maternal death reviews.

## Supporting information

**S1 File. Facility acuity definitions.**
(DOCX)

**S2 File. Characteristics of birthing sites included in the study in Argentina, Ghana, and India.**
(DOCX)

## Acknowledgments

The authors would like to thank the following people, without whose efforts the publication of this manuscript would not have been possible:

In Argentina, we gratefully acknowledge the support of the National Directorate of Maternal, Child and Adolescent Health and the Directorate of Sexual and Reproductive Health of the Ministry of Health of the Nation. We commend the compromise and dedication of the provincial teams, members of the Maternal and Child Health Programs of the Provincial Ministries of Health: Dr. Adriana Martirena, Dr. Daniel Nowacky, Dr. Adriana Allones, Marta Ferrary, Dr. Claudia Castro, Ana Seimande, Antonio Tabarcachi, Noelia Coria, Cintia Jacobi, Laura Soto, Dr. Mara Bazán, Dr. Susana Velazco, Dr. Patricia Leal, and Marcela Tapia. Finally, we would like to express our deepest gratitude to all of the health workers who participated in the study as data collectors, working through the height of the COVID-19 pandemic in Argentina.

In Ghana, we gratefully acknowledge the the support of the Director General–Ghana Health Service–Dr. Patrick Kuma-Aboagye and the Family Health Division of Ghana Health Service; Dr. Ernest Konadu Asiedu, Ms. Roberta Asiedu, Dr. Margretta Chandi and Ms. Catherine Adu Asare; Ms. Philomina Agbasi, Ms. Nyarko Agyemang-Duah, Mr. Tony Godi; and all regional and district health workers and field teams for their persistence in data collection and entry despite the challenges.

In India, we gratefully acknowledge the support of Dr. Dinesh Baswal, Ex Deputy Commissioner at Maternal Health Division, Ministry of Health & Family Welfare, India; the Mission Directors, State Health Departments of Tamil Nadu and Uttar Pradesh, and the District health Officials of study districts. We also acknowledge the support of Dr. Manju Chhugani and Dr. Renu Kharb for their guidance in review of the secondary data on many indicators. Finally, we sincerely thank the district field teams for their untiring efforts and adaptation to new methodologies to collect good quality data, in the midst of COVID in India. We also thank all the health workers and facility staff who participated in the study despite their busy schedules due to COVID situation.

## Author Contributions

**Conceptualization:** Jewel Gausman, Ernest Kenu, Richard Adanu, Mabel Berrueta, Ana Langer, Verónica Pingray, Niranjan Saggurti, Paula Vázquez.

**Data curation:** Jewel Gausman, Delia A. B. Bandoh, Suchandrima Chakraborty, Nizamuddin Khan, Carolina Nigri, Magdalene A. Odikro, Sowmya Ramesh, Caitlin R. Williams.

**Formal analysis:** Jewel Gausman, Delia A. B. Bandoh, Suchandrima Chakraborty, Nizamuddin Khan, Carolina Nigri, Magdalene A. Odikro, Sowmya Ramesh, Caitlin R. Williams.

**Funding acquisition:** Ana Langer.

**Methodology:** Jewel Gausman, Ernest Kenu, Richard Adanu, Mabel Berrueta, Suchandrima Chakraborty, Verónica Pingray, Sowmya Ramesh, Niranjan Saggurti, Paula Vázquez, Caitlin R. Williams, R. Rima Jolivet.

**Project administration:** Jewel Gausman, Mabel Berrueta, Ana Langer, Verónica Pingray, R. Rima Jolivet.

**Supervision:** Jewel Gausman, Ernest Kenu, Richard Adanu, R. Rima Jolivet.

**Visualization:** Jewel Gausman.

**Writing – original draft:** Jewel Gausman, R. Rima Jolivet.

**Writing – review & editing:** Jewel Gausman, Ernest Kenu, Richard Adanu, Delia A. B. Bandoh, Mabel Berrueta, Suchandrima Chakraborty, Nizamuddin Khan, Ana Langer, Carolina Nigri, Magdalene A. Odikro, Verónica Pingray, Sowmya Ramesh, Niranjan Saggurti, Paula Vázquez, Caitlin R. Williams, R. Rima Jolivet.

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
