## [Decision Letter · Decision Letter 0]

13 Apr 2023

PONE-D-22-28776Validating the indicator “maternal death review coverage” to improve maternal mortality data: A retrospective analysis of district, facility, and individual medical record dataPLOS ONE

Dear Dr. Gausman,

Thank you for submitting your manuscript to PLOS ONE. After careful consideration, we feel that it has merit but does not fully meet PLOS ONE’s publication criteria as it currently stands. Therefore, we invite you to submit a revised version of the manuscript that addresses the points raised during the review process.

We look forward to receiving your revised manuscript.

Kind regards,

George Kuryan

Academic Editor

PLOS ONE

2. Please ensure that you have specified (1) whether consent was informed and (2) what type you obtained (for instance, written or verbal, and if verbal, how it was documented and witnessed). If your study included minors, state whether you obtained consent from parents or guardians. If the need for consent was waived by the ethics committee, please include this information.

“This work was supported by the Bill and Melinda Gates Foundation through an award to RRJ and AL (Improving Maternal Health Measurement (IMHM) Capacity and Use, grant number OPP1169546). Funders had no role in study design, data collection and analysis, decision to publish, or preparation of the manuscript.”

“This work was supported by the Bill and Melinda Gates Foundation through an award to RRJ and AL (Improving Maternal Health Measurement (IMHM) Capacity and Use, grant number OPP1169546). Funders had no role in study design, data collection and analysis, decision to publish, or preparation of the manuscript.”

Additional Editor Comments:

I concur in large part with the reviewer

METHOD:

Study setting: The readers need more information here. Briefly describe why those countries were selected, whether the study is part of a larger study, location of those districts, either rural and / or urban. How the health system is basically organized in each country from lower to higher level health facility, what type of health workers can be found in each level of health facility (for instance: the requirement can be to have at least one nurse and midwife in each health post), etc.

Data collection and source: Why only public health facilities are included in India and Argentina? What is the data collection period? Tell us more about the expertise data collectors (are they health professionals? How the training was done? Etc.) what tools were used to collect the data, ect…

District/provincial data: Was individual level data or aggregate level data extracted from HMIS?

- If aggregate level, how the linkage was done with facility level data?

- If individual level, what variables were extracted to be used to link with facility data (only date of death is not sufficient in my view). You need to convince why only this information was enough to link facility and district level data.

Please be more precise.

Data analysis: You need to define your indicators (completeness, content, death with evidence, etc.). What criteria was used for each indicator? For instance, what is “death evidenced at facility” what criteria was used to help classify it as Yes or No.

Socio-demographic characteristics of women (age, place of residence and timing of death): Why only those indicators are selected. Is it based on literature review? How are they selected? Ar those only available variables collected in death certificate? Why not include additional variables such as profession, education, marital status, etc… if available.

Lines 208-211: We need more details about the linkage between facility and district level data. Variables used for this linkage. I do not think only date of death is enough to link two datasets. There are linkage procedures that were developed to link two datasets using names, place of death, date of death, etc. Be more specific about technic you have used here.

RESULTS:

There are too many tables and figures. Many of them can be combined or summarized. Readers may be lost here with too many repeated results. For instance the last table (table 6), almost all information were found in table 1.

While this has some merit the poor quality of the write up and analysis detract from The findings

Reviewers' comments:

Reviewer's Responses to Questions

**Comments to the Author**

1. Is the manuscript technically sound, and do the data support the conclusions?

Reviewer #1: Yes

2. Has the statistical analysis been performed appropriately and rigorously? 

Reviewer #1: Yes

3. Have the authors made all data underlying the findings in their manuscript fully available?

Reviewer #1: Yes

4. Is the manuscript presented in an intelligible fashion and written in standard English?

Reviewer #1: Yes

5. Review Comments to the Author

Reviewer #1: VALIDATING THE INDICATOR “MATERNAL DEATH REVIEW COVERAGE” TO IMPROVE MATERNAL MORTALITY DATA: A RETROSPECTIVE ANALYSIS OF DISTRICT, FACILITY, AND INDIVIDUAL MEDICAL RECORD DATA

Maternal mortality in LMIC remains is an important topic to explore due to high persistence of MMR even at a health facility level. The authors used an interesting approach to assess the maternal death review coverage. The results of the paper should guide policymarkers on the improvement of maternal mortality data and maternal health programs.

The Introduction is well written. But it is too long. Recommend to shorten it.

METHOD:

Study setting: Even if the authors asked to review the study protocol for more detail, the readers need more information here. For instance, I would suggest to briefly describe why those countries were selected, whether the study is part of a larger study, location of those districts, either rural and / or urban. How the health system is basically organized in each country from lower to higher level health facility, what type of health workers can be found in each level of health facility (for instance: the requirement can be to have at least one nurse and midwife in each health post), etc.

Data collection and source: Why only public health facilities are included in India and Argentina? What is the data collection period? Tell us more about the expertise data collectors (are they health professionals? How the training was done? Etc.) what tools were used to collect the data, ect…

District/provincial data: Was individual level data or aggregate level data extracted from HMIS?

- If aggregate level, how the linkage was done with facility level data?

- If individual level, what variables were extracted to be used to link with facility data (only date of death is not sufficient in my view). You need to convince why only this information was enough to link facility and district level data.

Please be more precise.

Data analysis: You need to define your indicators (completeness, content, death with evidence, etc.). What criteria was used for each indicator? For instance, what is “death evidenced at facility” what criteria was used to help classify it as Yes or No.

What is completeness of death review, what is the criteria? You may have mentioned it in the Box 1 (Introduction), but we still need it here.

Socio-demographic characteristics of women (age, place of residence and timing of death): Why only those indicators are selected. Is it based on literature review? How are they selected? Ar those only available variables collected in death certificate? Why not include additional variables such as profession, education, marital status, etc… if available.

Lines 208-211: We need more details about the linkage between facility and district level data. Variables used for this linkage. I do not think only date of death is enough to link two datasets. There are linkage procedures that were developed to link two datasets using names, place of death, date of death, etc. Be more specific about technic you have used here.

RESULTS:

There are too many tables and figures. Many of them can be combined or summarized. Readers may be lost here with too many repeated results. For instance the last table (table 6), almost all information were found in table 1.

I suggest reorganizing the results’ section to summarize your main findings in 3-4 tables and 1-2 graphs. Numbers from one table to another are different and confusing.

Lines 241-258: This section is difficult to read. You start with age group then residence area and talk again about age group. I suggest to reorganize it taking into account one characteristic at a time.

Minors

Line 49: correct “indictor” to “indicator”

Line 113-5: something is missing in this sentence, not clear

Lines 139-9: Not clear, do you say that death that occurred outside the health facility may be counted in the denominator. I thought this indicator only considers intra-facility (hospital) death.

Table 3. Ghana (Yes, column 6), correct n should be 14 instead of 13

Table 4. Review the total number for India.

Line 278: Review the sentence. I think it should be “higher than” instead of “lower than”

6. PLOS authors have the option to publish the peer review history of their article (what does this mean?). If published, this will include your full peer review and any attached files.

Reviewer #1: **Yes: **Almamy Malick Kante

---

## [Author Response · Author response to Decision Letter 0]

1 Jun 2023

We thank the editor and reviewer for their comments. We have uploaded a detailed point by point response to all comments.

---

## [Decision Letter · Decision Letter 1]

26 Jul 2023

PONE-D-22-28776R1Validating the indicator “maternal death review coverage” to improve maternal mortality data: A retrospective analysis of district, facility, and individual medical record dataPLOS ONE

Dear Dr. Gausman,

Thank you for submitting your manuscript to PLOS ONE. After careful consideration, we feel that it has merit but does not fully meet PLOS ONE’s publication criteria as it currently stands. Therefore, we invite you to submit a revised version of the manuscript that addresses the points raised during the review process.

We look forward to receiving your revised manuscript.

Kind regards,

George Kuryan

Academic Editor

PLOS ONE

Reviewers' comments:

Reviewer's Responses to Questions

**Comments to the Author**

1. If the authors have adequately addressed your comments raised in a previous round of review and you feel that this manuscript is now acceptable for publication, you may indicate that here to bypass the “Comments to the Author” section, enter your conflict of interest statement in the “Confidential to Editor” section, and submit your "Accept" recommendation.

Reviewer #2: (No Response)

2. Is the manuscript technically sound, and do the data support the conclusions?

Reviewer #2: Yes

3. Has the statistical analysis been performed appropriately and rigorously? 

Reviewer #2: Yes

4. Have the authors made all data underlying the findings in their manuscript fully available?

Reviewer #2: Yes

5. Is the manuscript presented in an intelligible fashion and written in standard English?

Reviewer #2: Yes

6. Review Comments to the Author

Reviewer #2: This study addressed an important area od validity and quality of maternal death statistics based on MDR. The study has clear approach relating to the indicators used globally and issues addressing both the numerator and denominator.

However, there are some concerns which need to be addressed.

1. While the authors make enormous efforts in six districts over three countries, at the end the number of deaths which from the core of the analyses are small. 17 in Argentina, 14 in Ghana and 58 in India. The numbers are too small to make generalizations even within the country. One cannot escape a sense that there is over-interpretation based on a small set of data.

2. While the introduction makes a good case for the need for the study, methods section is quite confusing. Overall, the paper fails to have an engaging narrative and that subtracts significantly from the importance of the paper. I would recommend a tabular approach to list the key questions /indicators and methodological approach to answering them.

3. The formation of tables, too many, needs rethinking. Since inter-country comparison is neither attempted nor necessary, could the authors consider using country specific tables with each indicator as a row. and provide a clear narrative of a country as a case study before going to next and maybe use summary table for comparison.

4. To fully understand the results and their implications to make the recommendation, it would be useful to add some observations related to the process of reviews and reporting made by the investigators.

5. Lack of inclusion of private facilities in India and Argentina is a serious limitation.

Minor:

• Table 3 shows that both in Argentina and India approx. 17% of maternal deaths, had no evidence that a death review occurred (L 325-26); the denominator for Fig1 is not clear, whether it excludes these.

• Table 4 says India complete death review is 48 whereas it is only 24. 48 deaths had any review and not complete review. Needs correction.

• Table 5 can be simplified by deleting all NO as it is a binary and except complicating the table, add no value to the paper.

• L204 – six weeks of abortion and 42 days of delivery ).. Aren’t these same then why use different time units.

• Table 1 can go as supplementary table.

• Quality of figures are very poor. Fig1 need to have numbers, a horizontal bar could be considered for ease in interpretation. Figure 2 summary is a good idea but was difficult to interpret.

7. PLOS authors have the option to publish the peer review history of their article (what does this mean?). If published, this will include your full peer review and any attached files.

Reviewer #2: **Yes: **Anand Krishnan

---

## [Author Response · Author response to Decision Letter 1]

12 Sep 2023

We have uploaded a file containing our detailed, point-by-point response to reviewers; however, as requested, we have also pasted them below. 

Reviewers' comments:

Reviewer's Responses to Questions

Comments to the Author

1. If the authors have adequately addressed your comments raised in a previous round of review and you feel that this manuscript is now acceptable for publication, you may indicate that here to bypass the “Comments to the Author” section, enter your conflict of interest statement in the “Confidential to Editor” section, and submit your "Accept" recommendation.

Reviewer #2: (No Response)

2. Is the manuscript technically sound, and do the data support the conclusions?

Reviewer #2: Yes

3. Has the statistical analysis been performed appropriately and rigorously? 

Reviewer #2: Yes

4. Have the authors made all data underlying the findings in their manuscript fully available?

Reviewer #2: Yes

5. Is the manuscript presented in an intelligible fashion and written in standard English?

Reviewer #2: Yes

6. Review Comments to the Author

Reviewer #2: This study addressed an important area od validity and quality of maternal death statistics based on MDR. The study has clear approach relating to the indicators used globally and issues addressing both the numerator and denominator.

However, there are some concerns which need to be addressed.

1. While the authors make enormous efforts in six districts over three countries, at the end the number of deaths which from the core of the analyses are small. 17 in Argentina, 14 in Ghana and 58 in India. The numbers are too small to make generalizations even within the country. One cannot escape a sense that there is over-interpretation based on a small set of data.

Response: We thank the reviewer for this comment; however, we would like to clarify to the reviewer that our study consists of a census of all maternal deaths that occurred within health facilities within a given geographical unit. We could not have a larger “n” because we included all maternal deaths that occurred within all facilities that report to the government within each geographical unit. We believe that the important takeaway of our study is that even though maternal death is rare and there are relatively few maternal deaths in these geographical areas, we still found important challenges with regard to data quality and completeness. We have made the wording more clear in the methods section to emphasize that we have included all maternal deaths (see for example, the revisions on page 12, line 295-296). It seems that this issue highlighted by the reviewer relates more to concerns over generalizability, rather than inference.

We further provide extensive discussion of the limitations of our data in the discussion section. Of note, we would like to highlight for the reviewer the following statement in the discussion section:

“This study is subject to several strengths and limitations. A strength of this study is that we verified all maternal deaths, maternal death reviews, and content of reviews by collecting primary data from birth facilities, and we cross-validated facility-level data with data reported to districts. In terms of limitations, the number of deaths included in our studies was quite small as maternal death is a rare event. Therefore, our ability to draw statistical inferences from our data is limited and we report numerical, qualitative differences only.” 

To which we have added based on the reviewer’s feedback:

“We can also only generalize to the geographic units in which our study took place, as national-level generalizability is not the purpose of this study.”

2. While the introduction makes a good case for the need for the study, methods section is quite confusing. Overall, the paper fails to have an engaging narrative and that subtracts significantly from the importance of the paper. I would recommend a tabular approach to list the key questions /indicators and methodological approach to answering them.

Response: We thank the reviewer for this feedback. We have made extensive revisions to the methods section to more clearly articulate the different ways we operationalized the numerator and denominator. We would like to point the reviewer to the following changes:

With regard to the numerator:

“To validate the indicator, we varied the indicator’s numerator and denominator and explored the differences in the indicator’s value. 

We began by validating the numerator’s indicator, which was defined as the number of maternal deaths that occurred at a facility that were reviewed. To validate the numerator, we operationalized it in three different ways using different sources of data:1) the number of maternal deaths that occurred in a facility and that had evidence of a review, 2) the number of maternal deaths that occurred in a facility and that had a review that met the WHO MPDSR standard, and 3) the number of maternal deaths with reviews that were reported to the district/provincial level.”

With regard to the denominator, we have added: 

“The indicator’s denominator is defined as all maternal deaths that occurred in a facility within a given geographical unit. To verify the denominator, we varied the data source from which the counts of facility-based maternal deaths were obtained. We derived the count of maternal deaths from two different sources 1) the total number of maternal deaths evidenced at health facilities, and 2) the total number maternal deaths reported to the district/provincial level.”

And finally, with regard to the overall construction of the indicator: 

“In the final step, we varied construction of the indicator to explore differences in its value. We calculated the indicator three different ways reflecting the different data sources available for both the numerator and denominator as detailed in Table 1.”

Table 1: Indicator Definitions for “Maternal Death Review Coverage”

 Indicator Definition 1 Indicator Definition 2 Indicator Definition 3

Numerator Number of maternal deaths documented at a facility that have evidence of a review Number of maternal deaths documented at a facility that have evidence of a complete review Number of maternal death reviews reported to district/province

Denominator Number of maternal deaths evidenced at facilities Number of maternal deaths evidenced at facilities Number of maternal deaths reported to district/province

Data Source Facility Facility HMIS

3. The formation of tables, too many, needs rethinking. Since inter-country comparison is neither attempted nor necessary, could the authors consider using country specific tables with each indicator as a row. and provide a clear narrative of a country as a case study before going to next and maybe use summary table for comparison.

Response: While we appreciate the reviewer’s suggestion, we respectfully disagree with this comment on the following basis. We believe that presenting each country separately would be repetitive given that there are many shared points derived from the interrogation of each of the indicator’s components across each study area. We think that it is important to highlight the variation in data quality across sites, which would be challenging if we presented as three individual case studies. We systematically interrogate the value of the numerator, then the denominator, then the full value of the indicator across each study. 

4. To fully understand the results and their implications to make the recommendation, it would be useful to add some observations related to the process of reviews and reporting made by the investigators.

Response: Again, we thank the reviewer for making suggestions to strengthen our paper, but we do not fully understand this comment. If the reviewer is referring to the process for collecting data on the content of death reviews, we respectfully point the reviewer to page 11 where we explain the process for abstracting data on death reviews in detail: 

“In all countries, facility data collected on death reviews included death date, review date, and type of documentation that supported the review (meeting minutes, meeting report, or other). We further reviewed the content and recorded whether the documentation available substantiated that the review satisfied the seven criteria outlined in the definitional standard (see Box 1). We sought to identify and use the most universally applicable and universally recognized reference standards available. Content was reviewed for compliance with the WHO technical guideline, this study’s reference standard. (9, 12, 35) Data collected from facilities at all sites used a standard paper-based data collection form with a separate question for each of the variables identified above. Each maternal death was documented individually on the form. One form was used for each facility.”

5. Lack of inclusion of private facilities in India and Argentina is a serious limitation.

Response: As detailed in the manuscript, we purposefully did not include private facilities in Argentina and India because private facilities in those countries do not report their data to the government, thus there is no HMIS data available for maternal deaths at those facilities. This is detailed on page 10, lines 195-198. We also noted this limitation in the discussion section of the version of the manuscript that was reviewed. 

To further address the reviewer’s comment, we added the following statement to the limitations noted in our manuscript “In Argentina and India, this is important given that private facilities were not included in the study, which could have an impact on both the numerator and the denominator as death reviews may be less frequent in facilities that are not part of the public hospital network; however, as private facilities in these countries do not report maternal deaths to the government, HMIS data were unavailable from such facilities, which would have limited our ability to compare the value of the indicator using different data sources.”

Minor:

• Table 3 shows that both in Argentina and India approx. 17% of maternal deaths, had no evidence that a death review occurred (L 325-26); the denominator for Fig1 is not clear, whether it excludes these. 

Response: We thank the reviewer for their close review. We have revised the sentence in the manuscript referring to Fig 1 as follows: “Audit findings detailing the content of each maternal death review conducted by definitional component across study countries are displayed in Fig 1.” Further, we have revised the axis label in the chart to read as follows: “Percent of Death Reviews Conducted Meeting Each Definitional Component.” 

• Table 4 says India complete death review is 48 whereas it is only 24. 48 deaths had any review and not complete review. Needs correction.

Response: We thank the reviewer for drawing our attention to this error. We have corrected the table. 

• Table 5 can be simplified by deleting all NO as it is a binary and except complicating the table, add no value to the paper.

Response: We have made the revision suggested by the reviewer and deleted the “no” columns. 

• L204 – six weeks of abortion and 42 days of delivery ).. Aren’t these same then why use different time units.

Response: We have taken the reviewer’s suggestion and changed 6 weeks to 42 days in relation to abortion for consistency. 

• Table 1 can go as supplementary table.

Response: We have taken the reviewer’s suggestion and moved Table 1 to supplementary material (now Table S2). 

• Quality of figures are very poor. Fig1 need to have numbers, a horizontal bar could be considered for ease in interpretation. Figure 2 summary is a good idea but was difficult to interpret.

Response: We are not sure what the reviewer is referring to by mention of a “horizontal bar” or “numbers” in relation to Fig 1, given that it is already presented as a horizontal bar chart. We have added additional tick marks on the horizontal axis in Figure 1 to support interpretation. For Fig 2, the purpose is simply to present and highlight subnational variability. We would like to direct the reviewer to the paragraph on page 27 beginning with line 548 where we provide an in-depth interpretation of the results presented in Fig 2. 

7. PLOS authors have the option to publish the peer review history of their article (what does this mean?). If published, this will include your full peer review and any attached files.

Do you want your identity to be public for this peer review? For information about this choice, including consent withdrawal, please see our Privacy Policy.

Reviewer #2: Yes: Anand Krishnan

---

## [Decision Letter · Decision Letter 2]

8 Feb 2024

PONE-D-22-28776R2Validating the indicator “maternal death review coverage” to improve maternal mortality data: A retrospective analysis of district, facility, and individual medical record dataPLOS ONE

Dear Dr. Gausman,

Thank you for submitting your manuscript to PLOS ONE. After careful consideration, we feel that it has merit but does not fully meet PLOS ONE’s publication criteria as it currently stands. Therefore, we invite you to submit a revised version of the manuscript that addresses the points raised during the review process.

We look forward to receiving your revised manuscript.

Kind regards,

Jianhong Zhou

Staff Editor

PLOS ONE

Journal Requirements:

Reviewers' comments:

Reviewer's Responses to Questions

**Comments to the Author**

1. If the authors have adequately addressed your comments raised in a previous round of review and you feel that this manuscript is now acceptable for publication, you may indicate that here to bypass the “Comments to the Author” section, enter your conflict of interest statement in the “Confidential to Editor” section, and submit your "Accept" recommendation.

Reviewer #2: (No Response)

2. Is the manuscript technically sound, and do the data support the conclusions?

Reviewer #2: Yes

3. Has the statistical analysis been performed appropriately and rigorously? 

Reviewer #2: Yes

4. Have the authors made all data underlying the findings in their manuscript fully available?

Reviewer #2: (No Response)

5. Is the manuscript presented in an intelligible fashion and written in standard English?

Reviewer #2: Yes

6. Review Comments to the Author

Reviewer #2: Overall the paper covers an important area and uses a sound approach.

Abstract:

• Use the term assessed rather than explore the validity of the indicator.

• Add the actual number of deaths covered in the study.

• Conclusions: First line can be deleted. It says we found incompleteness in deaths recorded. Is this result shown in the abstract?

Main Paper:

They selected on high performing and one low performing districts – are they reporting the results by this?

What proportion of the maternal deaths are expected to happen in the private sector needs to be told, even if this is approximate and is from the district level reporting.

I still think that there are too many tables and it is best to organize all indicators of one country in a table to give a composite idea of that country to make a meaningful interpretation. Repeated comparisons between countries (which is meaningless) in all tables hinders and is boring beyond a point. The common issues can then be taken up in discussion. The authors feel otherwise. I leave it for the editors to decide.

7. PLOS authors have the option to publish the peer review history of their article (what does this mean?). If published, this will include your full peer review and any attached files.

Reviewer #2: **Yes: **Anand Krishnan

---

## [Author Response · Author response to Decision Letter 2]

10 Apr 2024

Response to Reviewers

We thank the reviewer for a positive response on our manuscript and a decision of "minor revisions."

Reviewer #2: Overall the paper covers an important area and uses a sound approach.

Abstract:

• Use the term assessed rather than explore the validity of the indicator.

Response: We changed the term “explored” to “assessed” as follows: “This study assessed the validity of the indicator by examining both its numerator—the number and quality of death reviews—and denominator—the number of facility-based maternal deaths and comparing estimates of the indicator obtained from facility- versus district-level data.” (Page 2, line 51). 

• Add the actual number of deaths covered in the study.

Response: We added the following sentence to the abstract “In total, we found 17 deaths in Argentina, 14 deaths in Ghana, and 58 deaths in India evidenced at facilities.”

• Conclusions: First line can be deleted. It says we found incompleteness in deaths recorded. Is this result shown in the abstract?

Response: We changed the word “incompleteness…” to “we found discrepancies…” This is one of the main findings discussed in the abstract. 

Main Paper:

They selected on high performing and one low performing districts – are they reporting the results by this?

Response: We included this information for the reader to understand our selection process. We do not present any findings stratified by performance as we think that could be explored in a separate analysis and is beyond the scope of this paper. 

What proportion of the maternal deaths are expected to happen in the private sector needs to be told, even if this is approximate and is from the district level reporting. 

Response: This comment has been discussed extensively in previous versions. We cannot estimate with our data private sector deaths using district level data, as private facilities in India and Argentina do not report deaths to the government. Please see the limitations section for more detail, which reads: ““In Argentina and India, this is important given that private facilities were not included in the study, which could have an impact on both the numerator and the denominator as death reviews may be less frequent in facilities that are not part of the public hospital network; however, as private facilities in these countries do not report maternal deaths to the government, HMIS data were unavailable from such facilities, which would have limited our ability to compare the value of the indicator using different data sources.”

I still think that there are too many tables and it is best to organize all indicators of one country in a table to give a composite idea of that country to make a meaningful interpretation. Repeated comparisons between countries (which is meaningless) in all tables hinders and is boring beyond a point. The common issues can then be taken up in discussion. The authors feel otherwise. I leave it for the editors to decide.

Response: We strongly believe that the current format of the manuscript facilitates an iterative interrogation of the value of the indicator and data sources. We have worked extensively on the organization of this manuscript in order to make it clear and succinct. The purpose of the paper is to demonstrate the indicator’s validity in three countries, and it is not possible to pool the data, if that is what is recommended by the reviewer.

---

## [Editor Report · Decision Letter 3]

18 Apr 2024

Validating the indicator “maternal death review coverage” to improve maternal mortality data: A retrospective analysis of district, facility, and individual medical record data

PONE-D-22-28776R3

Dear Dr. Gausman,

We’re pleased to inform you that your manuscript has been judged scientifically suitable for publication and will be formally accepted for publication once it meets all outstanding technical requirements.

Kind regards,

Jianhong Zhou

Staff Editor

PLOS ONE
---

## [Editor Report · Acceptance letter]

7 May 2024

PONE-D-22-28776R3 

PLOS ONE

Dear Dr. Gausman, 

I'm pleased to inform you that your manuscript has been deemed suitable for publication in PLOS ONE. Congratulations! Your manuscript is now being handed over to our production team.

Kind regards, 

on behalf of

Dr. Jianhong Zhou 

Staff Editor

PLOS ONE